# Water Deficit Duration Affects Potato Plant Growth, Yield and Tuber Quality

Sheng Li *, Yulia Kupriyanovich, Cameron Wagg , Fangzhou Zheng and Sheldon Hann

Fredericton Research and Development Centre of Agriculture and Agri-Food Canada,
Fredericton, NB E3B 4Z7, Canada
* Correspondence: sheng.li@agr.gc.ca

**Abstract:** In humid climate regions, a short period of water deficit, especially during the vegetative growth and tuberization stages, has been found to affect potato plant growth, yield and tuber quality. However, there is still a lack of information on the impact of the water deficit duration. In this study, we examined potato plant growth, yield and tuber quality parameters with plants under 0 to 25 days of water deficit initiated at the beginnings of the vegetative growth stage and the tuberization stage, respectively. We found that for both the vegetative growth and tuberization stages, a longer water deficit duration resulted in no significant change in final plant height but significantly delayed flowering and reduced total biomass, yield, tuber dry matter content and share of large tubers. We estimate that per day of prolonged water deficit, there will be a yield loss of 3.1% and 3.4% for the vegetative growth and tuberization stages, respectively. Similarly, for per liter of irrigation water, there will be a yield increase of 16.3 g and 19.1 g for the vegetative growth and tuberization stages, respectively. Further studies are suggested to examine how supplemental irrigation can be used most effectively to mitigate the impact of water deficit on potato production in humid climate regions.

**Keywords:** drought; Russet Burbank; rain-fed agriculture; humid climate; supplemental irrigation



## 1. Introduction

Potato growth in cold, humid climate regions such as Atlantic Canada (AC) and the Northeast United States of America (USA) is predominantly under rain-fed agriculture [1]. Short periods of water deficit and water excess due to weather variations have always been a challenge for potato growers and have been recognized as a major contributing factor to yield loss in these regions [2,3]. With climate change, temperatures will rise and extreme weather events are expected to occur more frequently in the future [4]. As a result, short periods of water stress, especially water deficit, will be more severe and occur more frequently in the future. However, research on the water needs of potato crops has mostly been carried out in semi-arid and arid regions where irrigation is a prerequisite for potato cropping [3,5,6]. Results from the semi-arid and arid regions are not directly applicable to the humid climate regions due to differences in natural conditions and crop management [5,7,8]. Overall, for humid climate regions, there is a lack of quantitative information on the effects of short-period water stress on potato plant growth and its implications on potato yield and tuber quality.

There are a few studies in humid climate regions on the effects of continuous water deficit for the entire growing season on potato plant growth, yield and tuber quality (e.g., [7,8]). For short periods of water stress, the situation is more complicated and several aspects need to be considered. First, since water stress only occurs for short periods, we need to know the time when the potato plant is most sensitive to water stress. However, the few available studies on this topic are either affected by natural precipitation or the period studied is not short enough, making it hard to quantify the effects [3,9–11]. To answer this question, Wagg et al. conducted a pot experiment to test the effects of a short period

(two weeks) of water deficit and water excess, respectively, on potato plant growth, yield and tuber quality [12]. Four different potato growth stages (PGSs), sprouting, vegetative growth, tuberization and bulking, were examined. The results show that a short period of water deficit overall had negative impacts, while a short period of water excess had slightly positive impacts on potato plant growth, yield and quality. For water deficit, the most sensitive periods were the vegetative growth and tuberization stages, whereas for water excess, there was mostly no significant difference between different PGSs. These results confirmed the benefits of supplemental irrigation and suggest a low risk of over-irrigation in well-drained fields. They further suggested that when the availability of water for irrigation is limited, using water strategically based on potato plant water demands can maximize the benefits of supplemental irrigation.

The second aspect of short-period water stress is the duration of the period. The questions that need to be answered are: how long can the potato plant endure water stress? Will the potato plant be able to tolerate a certain length of time under a water deficit without suffering losses in potato plant growth, yield and tuber quality? If yes, how long is the duration, and if not, how much will the losses be with every day of water deficit? To our knowledge, there is no study on the direct effects of water deficit duration on potato plant growth, yield or tuber quality. In the previously mentioned study carried out by Wagg et al. [12], the duration of water stress was set the same for all treatments, so it is impossible to examine the effects of the water deficit duration. There has been some previous work on water-saving irrigation or deficit irrigation, which examined the impacts of different degrees of water deficit on potato growth and yield. A general conclusion from these studies is that there is a potential for yield loss or quality degradation with water-saving irrigation or deficit irrigation compared to full irrigation [13–16]. However, when the water supply is kept at a certain level (e.g., lower than the optimum level but still meets basic crop needs), the loss may be non-significant or the cost-saving may outweigh the yield and quality losses economically [17]. In these studies, the degrees of water deficit were often set at constant levels based on soil moisture (a level less than available water capacity) or irrigation (a level less than full irrigation). This fits the conditions in the semi-arid and arid climate regions, where there is almost always a water deficit for growing potatoes. However, this again does not fit the humid climate regions where water deficit only occur for short periods. The questions regarding the effects of water deficit duration remain unanswered.

In this study, we extended the experiment conducted in the previous study [12] to examine the effects of water deficit duration on potato plant growth, yield and tuber quality. We focused on the vegetative growth stage and the tuberization stage since these two stages were found to be the most sensitive to water deficit in our previous study. Our objective is to provide quantitative information on the responses in potato plant growth, yield and tuber quality to every day of prolonged water deficit during these two potato growth stages in order to develop water management strategies that fit the climate conditions and crop production systems in the cold, humid climate regions of North America.

## 2. Materials and Methods

### 2.1. Experimental Design

The experiment was conducted with pots in a controlled greenhouse at Agriculture and Agri-Food Canada's (AAFC) Fredericton Research and Development Centre (FRDC) in Fredericton, New Brunswick (NB), Canada. There were a total of 53 pots. Three pots were under control treatment, for which the potato was grown under a no-water-stress condition throughout the entire experimental period. The other 50 pots were divided into two groups of 25 pots, one group for the vegetative growth stage (VS) and the other group for the tuberization stage (TS). Within each group, each pot was under a specific duration of water deficit treatment ranging from 1 to 25 days. The initial water deficit phase began at 15 and 29 days after planting (DAP) for the VS and TS, respectively. Outside the allotted water deficit treatment period, the potato plants received a regular watering regime and were

kept under a no-water-stress condition. It should be noted that, with the exception of the control treatment, there was no replicate for any specific water deficit duration treatment (1 to 25 days), as here we were interested in assessing the changes in potato plant growth, yield and tuber quality with the number of days of water deficit over time, not the effects of fixed numbers of days of water deficit.

## 2.2. Experimental Conditions

The experiment was conducted using plastic pots measuring 29 cm in diameter. The pots were placed on a bench in the middle of the greenhouse, and the locations of the pots on the bench were randomized. The greenhouse was set to a 16 h day photoperiod, with the daytime and nighttime temperatures controlled in the ranges of 20–24 °C and 18–22 °C, respectively, which are similar to the natural light and temperature conditions in the growing season in NB. The soil used in the experiment was top soil taken from a field in NB. The soil texture was sandy loam with a sand, silt and clay content of 65%, 27% and 9%, respectively. The soil was sterilized and sieved through a 5 mm sieve. A scale was used to weigh 14 kg of soil, which was mixed with 6.6 g of 17-17-17 (NPK) granular fertilizer and then filled into the pot. The soil was packed to a target soil bulk density of 1.38 g cm$^{-3}$. One seed piece (55–63 g/seed piece) of Elite II Russet Burbank potato with at least 3 good eyes from the apical end of the seed piece was added to each pot. The Russet Burbank variety was used because it is the most popular variety used in NB and worldwide. All pots received 550 mL of 10-52-10 (NPK) starter fertilizer to promote growth.

Soil water retention was analyzed using the pressure extraction method (see details in [12]). The soil volumetric water content (SVWC) at field capacity (FC) and permanent wilting point (PWP) were 26.6% and 12.9%, respectively. The plant's available water capacity (AWC) was calculated as FC minus PWP, which was 13.6%. The SVMC was monitored daily (detailed method description in the following section) and was used as the basis to calculate the water volume required for irrigation. The SVWC used to trigger irrigation for the no-water-stress condition and the water deficit condition were set at 70% of the AWC (calculated to be 22.5%) and the PWP (i.e., 12.9%), respectively. In practice, for pots under the no-water-stress treatment, if the measured SVWC was greater than 22.5%, no water was added and if it was below 22.5%, the amount of water added was calculated so that the SVWC after adding the water was at about the level of FC (i.e., 26.6%). For pots under the water deficit treatment, if the measured SVWC was greater than the PWP (i.e., 12.9%), no water was added and if it was below 12.9%, the amount of water added was calculated so that the SVWC after adding the water was at a level slightly above the PWP.

## 2.3. Data Collection

Soil moisture and temperature were measured daily, approximately at the same time (10:00 a.m.), using a WET sensor (Delta-T Devices, Burwell, UK). Three readings were taken from the surface soil at approximately 10 cm depth and the average value was calculated and recorded as the SVWC and soil temperature for that day. These data were used to calculate the amount of water required to maintain the designed treatment as described above. Watering was performed manually after collecting all plant data, and the amount of water received by each plant was also recorded.

Potato plant emergence was checked daily until all potato sprouts emerged from the soil. After plant emergence, the plant height was measured with a meter stick for the longest branch from the stem base at the soil level to the apical bud. Plant height data were only recorded after the plant reached 10 cm in height. The stem count was completed daily after emergence until the number of stems stopped changing. Flowering was checked daily after plants reached the tuberization stage (four weeks after planting). The first day with a fully open flower was recorded as the flowering date. Note that some plants formed flowers before the designed water deficit period started, but the flowers did not open and dried out because of water stress when they were under the water deficit treatment. Most

of these plants were able to form flowers, which will then open later during the recovery period and this subsequent date was recorded as the flowering date.

A leaf porometer (Decagon Devices Inc., Pullman, WA, USA) was used to measure the stomatal conductance of the plant leaves. Since the measurement was time consuming, it was conducted only for selected experimental units (i.e., pots) as follows. First, two pots under the control treatment (no-water-stress throughout the experiment period) were measured daily, starting from the vegetative growth stage (15 days after planting). Second, the two pots that received the longest water deficit treatment for the vegetative growth stage and tuberization stage (VS-25 and TS-25), respectively, were measured daily during the period when the plant was under the water deficit condition. Last, for all other pots under the water deficit treatment, the measurement was only conducted at the end of the water deficit period. During the measurement, three readings were taken for the top branch leaf at approximately the mid-height plant level and branches from different sides of the plant were picked. The measured stomatal conductance was expressed in the unit of mmol m$^{-2}$ s$^{-1}$. The SPAD chlorophyll meter (Spectrum Technologies Inc., Aurora, IL, USA) was used to measure the 'greenness' of plants, which can be used as a biological indicator reflecting the plant's response to water deficit stress [18]. The measurement was conducted daily after the first leaf was formed. Three measurements were performed, and the average value was recorded. For consistency, all measurements were conducted on the terminal leaves of the third full-sized branch, starting from the top of the plant.

Potato yield and quality data were collected after the potato plant was harvested 74 days after planting. All vines were cut at the root collar level. Roots and tubers were separated and washed. Tubers were graded and counted. Fresh weights for the above-ground portion (vegetative biomass), roots and tubers were recorded separately. Plant materials were then dried at 55 °C. Dry weight was recorded and dry matter content (DM) was calculated as:

$$DM = (Dry\ Weight/Fresh\ Weight) \times 100\%$$

Regression analysis was used to examine the effects of water deficit duration on plant growth, yield and tuber quality parameters.

### 2.4. Statistical Analyses

The effects of water deficit duration on the measured potato plant growth, yield and tuber quality parameters were analyzed using linear regression. The analyses were performed for the two stages separately. Since many of the measured parameters were correlated with each other (Table S1), a redundancy analysis (RDA) was conducted to summarize variable inter-correlations and determine the underlying cause for these inter-correlations. The water deficit treatment and associated irrigation and soil condition parameters were used as independent variables, and crop growth, yield and tuber quality parameters were used as dependent variables in the RDA. The regression analyses and RDA were performed in R (version 4.3.1) with package 'vegan'.

## 3. Results

### 3.1. Soil Volumetric Water Content and Irrigation Water Usage

For the three control experiment units (pots), which were kept under no-water-stress conditions throughout the experimental period, the measured soil volumetric water content (SVWC) ranged from 14.0% to 32.5% with average values of 22.2%, 22.2%, and 21.4% (CV ranged from 12% to 16%), which were close to the target level of 22.5% (Figure 1a). For pots under water deficit treatments, SVWC during the designed water deficit periods ranged from 7.9% to 20.8%, with an average value of 12.2% (CV = 18.5%). This was reflected in the data for both the vegetative growth stage (VS) and tuberization stage (TS) groups, as both the compiled data from different pots under the water deficit treatment of different durations (VS-Compile and TS-Compile) and the data for the two pots with the longest water deficit periods (VS-25 and TS-25) showed SVWC values close to the target level of 12.9% (Figure 1a). High SVWC during the water deficit period occurred only in the first

three days for the pots of the VS group. This was likely due to the fact that the plants were small at this time, with low evapotranspiration and, therefore, it took days for the soil to dry down to the PWP level.

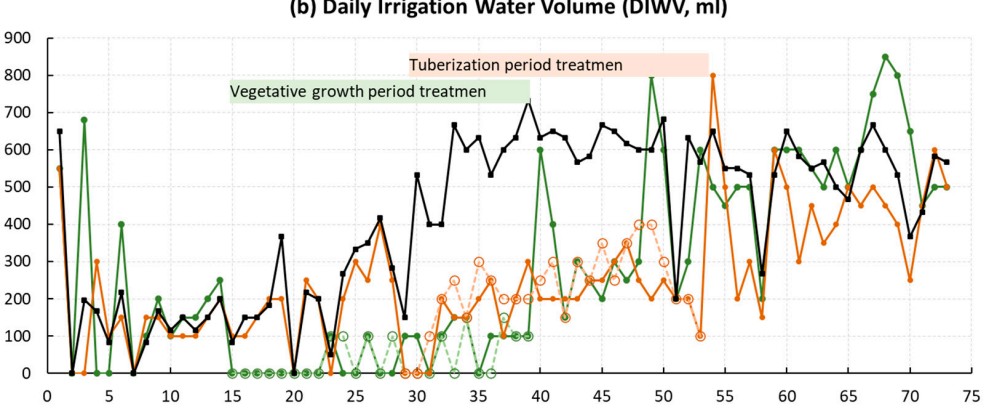

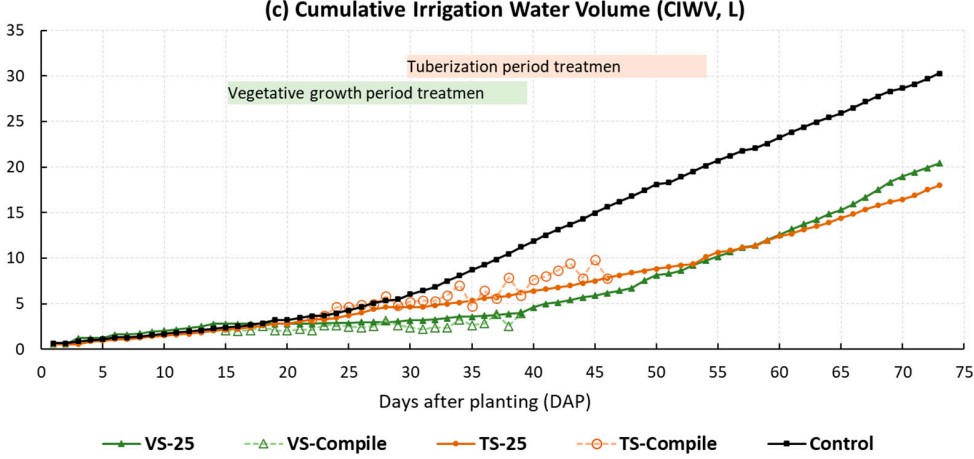

**Figure 1.** Daily soil volumetric water content and the amount of water used for selected experimental units (pots). VS-25 and TS-25 are the experimental units with the longest water deficit period, starting from the beginnings of the vegetative growth and tuberization stages, respectively. VS-Compile and TS-Compile are the compiled datasets for experimental units at the last days of their water deficit treatment, starting from the beginnings of the vegetative growth and tuberization stages, respectively. Control is the experimental unit under the control treatment (no-water-stress throughout the entire experiment period).

The daily irrigation water volume (DIWV) was calculated based on the SVWC measurements, so it varied with the SVWC values. For example, for the control pots, the big drops of irrigation occurred on day-29, day-51 and day-58, when the measured SVWC values were high (Figure 1a,b). Due to the low frequency of irrigation (i.e., daily, not hourly, or continuously) and errors associated with the soil moisture sensor, there were large day to day variations in the DIWV values. However, over the course of the experimental period, there was a general trend that the DIWV value varied with the growth of the potato plant. For example, for the control pots, the DIWV value was low from seeding to approximately day-20 because the plant had not been geminated or was very small and as such, the loss of water was mostly due to evaporation from the soil surface, whereas transpiration from the plant was negligible (Figure 1b). From day-20 to day-33, the DIWV value increased almost linearly, which likely was a reflection of the rapid vegetative growth during this period. The DIWV value was maintained at the same high level until approximately day-50, when the plant entered the tuber bulking stage and its leaves started to senesce. Then, the DIWV started to decline slightly until the end of the experiment. When plants were under the water deficit treatments, the DIWV values were low, which was a reflection of the designed experimental treatments (Figure 1b). After the treatment periods ended, the DIWV values did not recover immediately to the same levels as the control pots. Instead, they increased slowly. This was probably due to the impeded plant growth after the water deficit treatments, which took time for the plants to fully regain all their functional activities.

The cumulative irrigation water volume (CIWV) reflected the overall water stress levels for the treatments. There were large gaps between the curves for the control plants and those under the water deficit treatment (Figure 1c). It is interesting to note that the CIWV value for the VS-25 pot was lower than that for the TS-25 pot at first, but it increased faster after the water deficit periods ended and surpassed that for the TS-25 pot in the end. This is likely a reflection of the recovery of plant growth after the water deficit period ended, which was faster for VS than TS.

*3.2. Plant Growth*

The standard to which we compared plant performance was set by the control plants. These plants reached a height of 10 cm at day-20, and it increased fast from about day-22 to day-50, indicating accelerated shoot elongation. After day-50, the shoot elongation reached its maximum, so the rate of increase reduced substantially (Figure 2a). For the VS-25 pot (plant under water deficit treatment for 25 days since the beginning of the vegetative growth stage), the plant height was below 10 cm during the water deficit period, much lower than that of the control pots, indicating that the water deficit has significantly restricted plant growth. A similar effect was observed for the VS-Compile data (data compiled from pots with different durations of water deficit period since the beginning of the vegetative growth stage), for which the plant height stayed close to 10 cm. However, after the water deficit period ended, the plant height for the VS-25 pot picked up quickly and increased at a rate greater than that for the control pots and in the end, the plant height caught up with that of the control pots. For the TS-25 pot (plant was under water deficit treatment for 25 days since the beginning of the tuberization stage), the plant height was substantially lower than that of the control pots since the water deficit period started (Figure 2a). Similar to that of the VS-25, the plant in the TS-25 did recover after the water deficit period ended, but the rate of recovery was not as high as that for the VS-25 and the plant height did not catch up with that of the control pots in the end. The TS-Compile plant height data varied a lot but were close to those for the TS-25 and did show large differences from those of the control pots during the water deficit period. It should be noted that there were large variations in plant height from pot to pot. However, the overall trend was that for both the VS and TS treatments, plant height increase was restricted during the water deficit periods but recovered after the water deficit periods ended. The recovery of plant height under the VS treatment was stronger than that under the TS treatment, and in many cases, the final plant height under the VS treatment was greater than that of the control pots.

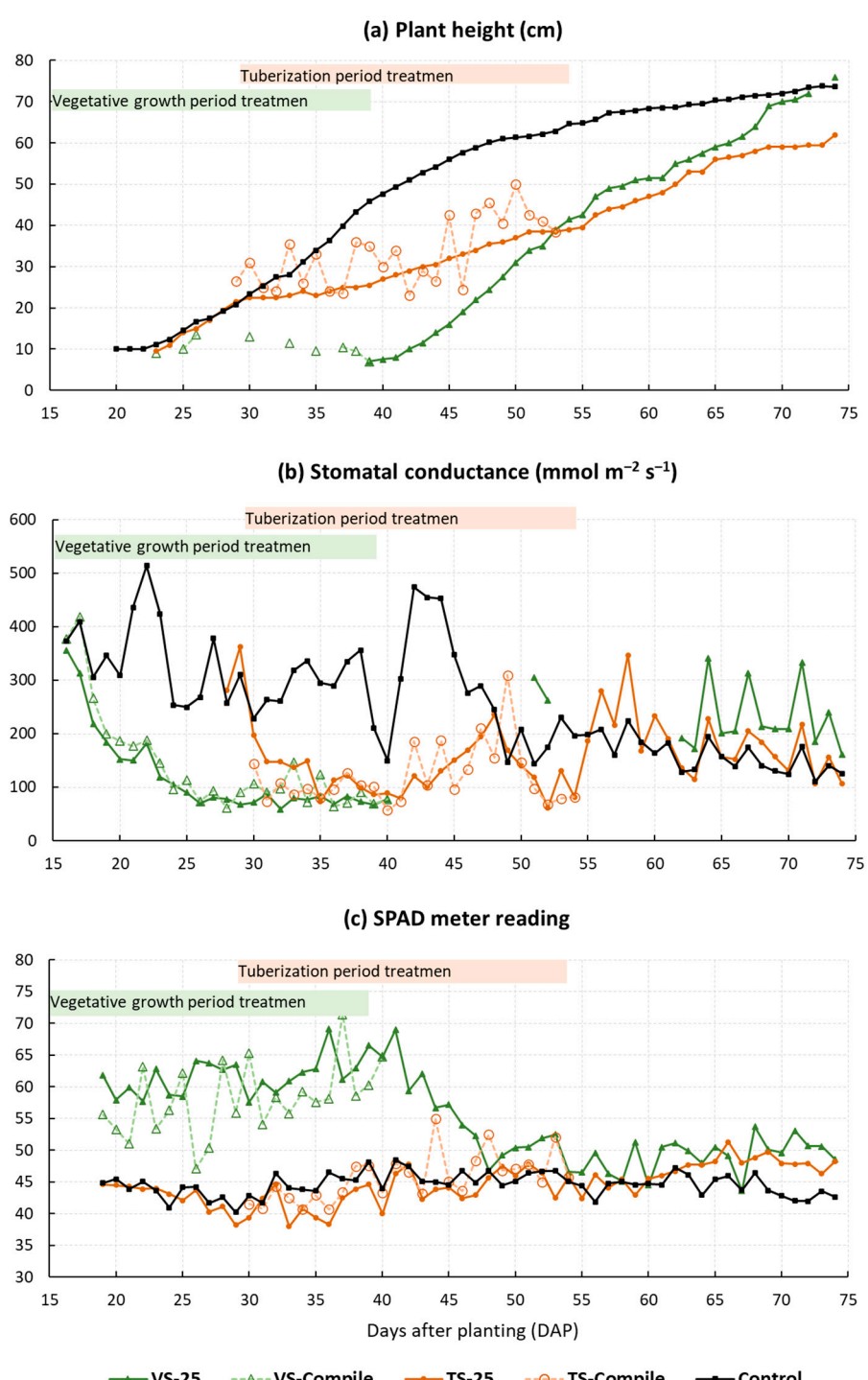

**Figure 2.** Plot height (**a**), stomatal conductance (**b**) and SPAD meter reading (**c**) for selected experimental units (pots). VS-25 and TS-25 are the experimental units with the longest water deficit period, starting from the beginnings of the vegetative growth and tuberization stages, respectively. VS-Compile and TS-Compile are the compiled datasets for experimental units at the last days of their water deficit treatment, starting from the beginnings of the vegetative growth and tuberization stages, respectively. Control is the experimental unit under the control treatment (no-water-stress throughout the entire experiment period).

The trend described above was also reflected in the regression analyses of the plant height versus the water deficit duration (Table 1). For example, at day-40, the day after the water deficit period ended for VS-25 (the pot with the longest duration of water deficit)

under the VS treatment. There was a significant correlation between plant height and the water deficit duration ($R^2$ = 0.75 ***). At day-54 (two weeks after the water deficit period ended for the VS treatment), the significance level of the correlation between plant height and the water deficit duration decreased ($R^2$ = 0.34 **) and the absolute value of the regression coefficient also decreased from 1.65 cm day$^{-1}$ to 0.64 cm day$^{-1}$, both of which can be attributed to the fact that the recovery of plant height has reduced the effect of the water deficit treatment. The effect of water deficit treatment continued to decrease with time and at day-70, the correlation between plant height and the water deficit duration was no longer significant. A similar trend can be observed for the TS treatment.

**Table 1.** Linear regression analysis between water deficit duration and potato plant growth, yield, tuber quality and soil water condition factors.

| Factor | | Vegetative Stage | | | Tuberization Stage | | |
|---|---|---|---|---|---|---|---|
| | | Slope | Intercept | $R^2$ | Slope | Intercept | $R^2$ |
| $H_{40}$ | 40-day plant height (cm) | −1.65 | 55.9 | 0.75 *** | −0.83 | 45.7 | 0.50 *** |
| $H_{54}$ | 54-day plant height (cm) | −0.64 | 70.8 | 0.34 *** | −0.75 | 65.5 | 0.57 *** |
| $H_{70}$ | 70-day plant height (cm) | 0.02 | 75.7 | 0.00 | −0.20 | 73.5 | 0.07 |
| D2F | Days to flowering (day) | 0.45 | 45.3 | 0.38 *** | 0.36 | 46.2 | 0.35 *** |
| SC | Stomatal conductance (mmol m$^{-2}$ s$^{-1}$) | −9.39 | 261.5 | 0.61 *** | −0.66 | 138.2 | 0.01 |
| SPAD | SPAD meter reading | 0.63 | 43.7 | 0.79 *** | 0.05 | 44.4 | 0.04 |
| $DW_V$ | Vegetation dry weight (g) | −0.20 | 46.2 | 0.15 * | −0.10 | 46.6 | 0.03 |
| $DW_R$ | Roots dry weight (g) | −0.06 | 7.0 | 0.25 * | 0.00 | 6.9 | 0.00 |
| TBM | Total biomass (g) | −2.41 | 118.0 | 0.76 *** | −2.15 | 109.8 | 0.76 *** |
| Y | Potato yield (tuber fresh weight, g) | −8.77 | 280.0 | 0.74 *** | −8.11 | 240.2 | 0.66 *** |
| $DW_T$ | Tuber dry weight (g) | −2.15 | 64.7 | 0.75 *** | −2.05 | 56.3 | 0.69 *** |
| TDMC | Tuber dry matter content (%) | −0.20 | 23.9 | 0.51 *** | −0.20 | 23.7 | 0.67 *** |
| $LT_n$ | Share of large tubers (>35 mm) by tuber number (%) | −2.04 | 45.2 | 0.65 *** | −1.70 | 44.9 | 0.41 *** |
| $LT_w$ | Share of large tubers (>35 mm) by fresh weight (%) | −3.59 | 83.1 | 0.72 *** | −3.03 | 86.6 | 0.46 *** |
| SVWC | Soil volumetric water content (%) | −0.36 | 21.4 | 0.95 *** | −0.41 | 22.2 | 0.97 *** |
| SVIW | Volume of irrigation water during the growth stage of interest (L) | −0.35 | 9.4 | 0.90 *** | −0.31 | 12.9 | 0.84 *** |
| ST | Soil temperature (°C) | 0.02 | 20.1 | 0.14 | −0.01 | 20.5 | 0.05 |

\* $p \leq 0.05$; *** $p \leq 0.001$.

Stomatal conductance for the control pots decreased with time throughout the experimental period from approximately 400 to 125 mmol m$^{-2}$ s$^{-1}$ (Figure 2b). For the VS treatment, both the VS-25 and VS-Compile data showed that the stomatal conductance dropped sharply at first when the water deficit started and then stabilized at approximately 85 mmol m$^{-2}$ s$^{-1}$ after about ten days of the water deficit. This indicates that the stomatal conductance had an immediate response to the water deficit. Data were missing after day-40 for VS-25 so it was not possible to know how the stomatal conductance changed immediately after the water deficit period ended. However, based on data from day-62 to day-74, the stomatal conductance appeared to have returned to a level slightly higher than that of the control. For the TS treatment, the stomatal conductance also dropped sharply at first, then stabilized at approximately 100 mmol m$^{-2}$ s$^{-1}$ after about eight days of water deficit, except for a small peak between day-45 and day-50. The TS-25 data also showed that there was an immediate recovery after the water deficit period ended. For the

rest of the experiment period, the stomatal conductance of TS-25 was very close to that of the control.

The SPAD meter reading for the control plants remained between 40 and 50 throughout the experiment period (Figure 2c). For the VS-25 plant, the SPAD meter reading increased to between 55 and 70 during the water deficit period (day-15 to day-39) and then dropped down slowly until day-49 and was maintained between 45 and 55 afterwards, slightly higher than that of the control. The VS-Compile data also showed a higher SPAD meter reading than that of the control during the water deficit period, although the variation was greater than that of VS-25. A completely different pattern was observed for the TS-25 plant and the TS-Compile data: the SPAD meter reading was similar to that of the control throughout the experimental period. Overall, the SPAD meter reading data suggest that the greenness or chlorophyll content of the leaves was significantly increased under the VS water deficit treatment but was not affected under the TS water deficit treatment.

For the days-to-flowering measurement, it increased significantly with the water deficit duration under both the VS and TS treatments ($R^2$ = 0.35 ** and 0.38 ***, respectively; Table 1). The regression analysis suggests that with every day of prolonged water deficit, the flowing date was delayed by 0.45 and 0.36 days for the VS and TS treatments, respectively.

### 3.3. Plant Biomass

For plant biomass, the regression analyses suggest that the vegetation dry weight decreased significantly with the water deficit duration under the VS treatment ($R^2$ = 0.15 *) but was not significantly affected by it under the TS treatment ($R^2$ = 0.03$^{NS}$, Table 1). Similarly, root dry weight decreased significantly with the water deficit duration under the VS treatment ($R^2$ = 0.25 **) but not under the TS treatment ($R^2$ = 0.00$^{NS}$). However, for tuber dry weight, it decreased significantly with the water deficit duration under both the VS and TS treatments ($R^2$ = 0.75 *** and 0.69 ***, respectively). The total biomass followed a similar pattern as that for tuber dry weight. The plant biomass results seem to contradict the results of plant height. Recall that the plant height recovered after the water deficit period ended under both the VS and TS treatments, and the water deficit duration did not show a significant effect on plant height in the end. This suggests that plant height may not be a good indicator for biomass accumulation.

### 3.4. Tuber Characteristics

Potato yield (i.e., tuber fresh weight) decreased significantly with the water deficit duration under both the VS and TS treatments ($R^2$ = 0.74 *** and 0.66 ***, respectively; Table 1). Based on the regression analyses, with every day of prolonged water deficit, the yield decreased by 8.8 g under the VS treatment. This was equivalent to a 3.1% yield loss per day of water deficit, calculated based on the intercept value (279.9 g), representing the yield under the control treatment. Similarly, for the TS treatment, yield decreased by 8.1 g with every day of prolonged water deficit, which was equivalent to a 3.4% yield loss per day. Tuber dry matter content also decreased significantly with the water deficit duration by 0.20% and 0.20% per day, respectively, under the VS and TS treatments ($R^2$ = 0.51 *** and 0.67 ***, respectively). In addition, shares of numbers and weight of large tubers (>35 mm in diameter) both decreased under water deficit conditions. For every day of prolonged water deficit, under the VS treatment, the shares of numbers and weight of large tubers decreased by 2.0% and 3.6%, respectively, whereas under the TS treatment, they decreased by 1.7% and 3.0%, respectively.

### 3.5. The Effects of Total Volume of Irrigation Water

Logically, the experimental unit with a longer water deficit duration should receive less total volume of irrigation water (TVIW). However, this is not always the case in this study due to the differences among individual plants as well as errors associated with the SVWC measurements. However, the trend is still for lower TVIW with longer water deficit duration and the correlation between water deficit duration and the TVIW was

statistically significant ($R^2$ = 0.61 ***) with a regression coefficient of −0.34 L day$^{-1}$. This can be interpreted as meaning that in our experiment, with one more day of water deficit, the total volume of irrigation water has been reduced by 0.34 L on average.

Since the TVIW was significantly correlated with the water deficit duration and potato plant growth and tuber development were strongly affected by water deficit duration, it was not a surprise to find that many plant biomass and tuber characteristic measures were also significantly correlated with the TVIW (Figure 3). The regression analyses suggest that with per liter of irrigation water: total biomass increased by 4.5 g and 5.1 g ($R^2$ = 0.84 *** and 0.79 ***), yield increased by 16.3 g and 19.1 g ($R^2$ = 0.78 *** and 0.68 ***), tuber dry matter content increased by 0.41% and 0.47% ($R^2$ = 0.65 *** and 0.68 ***), and the share by fresh weight of >35 mm tubers increased by 5.9% and 5.7% ($R^2$ = 0.58 *** and 0.31 **) under the VS and TS treatments, respectively. In particular, the increase in yield per unit volume of irrigation water is often defined as irrigation water use efficiency (IWUE). Our results are in a similar range to those reported by other researchers [19].

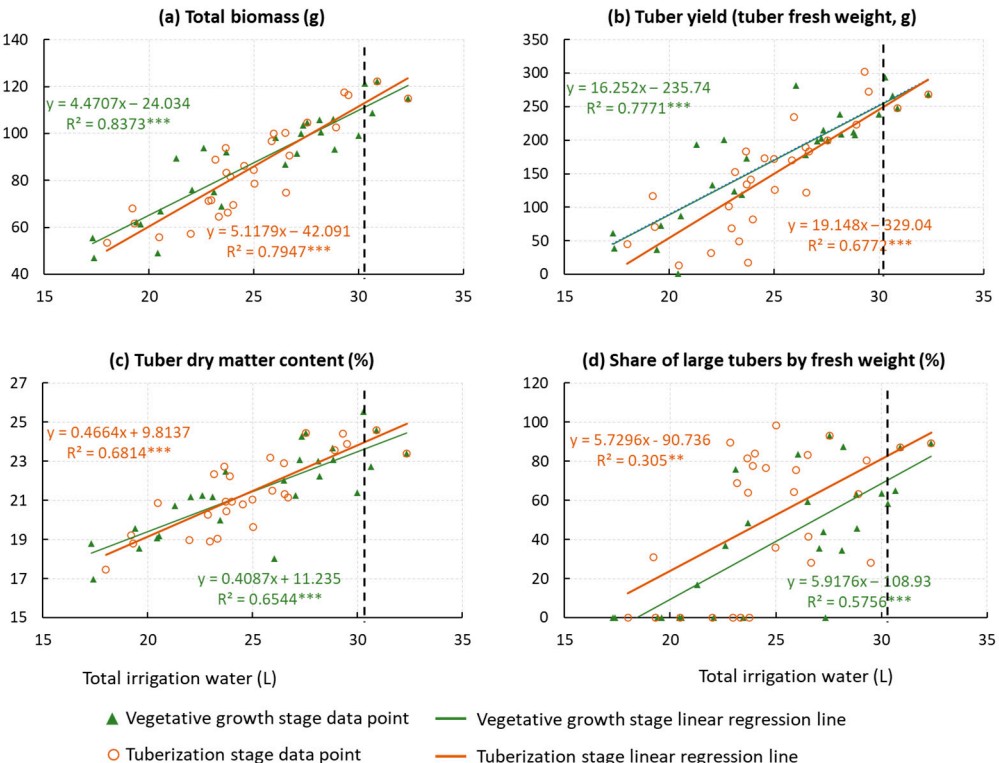

**Figure 3.** Variations of potato biomass and tuber characteristic parameters with the total amount of irrigation water. Vertical dashed black line denotes the total volume of irrigation water under the control (i.e., no water deficit) treatment (** $p \leq 0.01$; *** $p \leq 0.001$).

## 4. Discussion

### 4.1. The Common Effects and Differences between VS and TS

This study confirms that water deficit have significant negative impacts on potato plant growth and tuber development, as reported in many previous studies [7,12,20,21]. For both the vegetative growth and tuberization stages, the following trends were observed from Figures 1 and 2: (1) when the potato plant was under water deficit treatment, shoot elongation was significantly reduced; (2) after the water deficit period ended, the plants started to recover and in the end, plant heights for those under water deficit treatments were similar to those of the control; (3) the biomass accumulation for plants under water deficit treatments did not recover after the water deficit period ended and was reflected the most on the tubers, with those under water deficit treatments having significantly lower tuber dry mass than those of the control; and (4) the potato plants under water deficit treatments tended to produce smaller tubers. These trends are also well reflected

in the RDA biplot (Figure 4). The vector of $H_{40}$ aligns well with the first axis but is in the opposite direction as the vector of water deficit duration, indicating water deficit reduced vegetative growth at day-40. However, $H_{54}$ and $H_{70}$ no longer align well with the first axis, indicating that the water deficit did not have much effect on vegetative growth on day-56 and day-70 because the plant recovered after the water deficit ended. In the same vein, the Y, $DW_T$, TDMC, SC, $LT_n$ and $LT_w$ vectors are all opposite to the water deficit duration vector, indicating the negative impact of the water deficit on potato yield and tuber quality. Overall, these results suggest that when the potato plant is in a water deficit, its vegetative growth will be reduced. After the drying period ends, the plant will regain vegetative growth, but the ability of primary production and accumulation of biomass are still compromised and such compromise has long-lasting effects, leading to smaller tubers with a lower dry matter content and thus a lower yield than the control. Similar results have been reported in a previous study conducted by the same group [12].

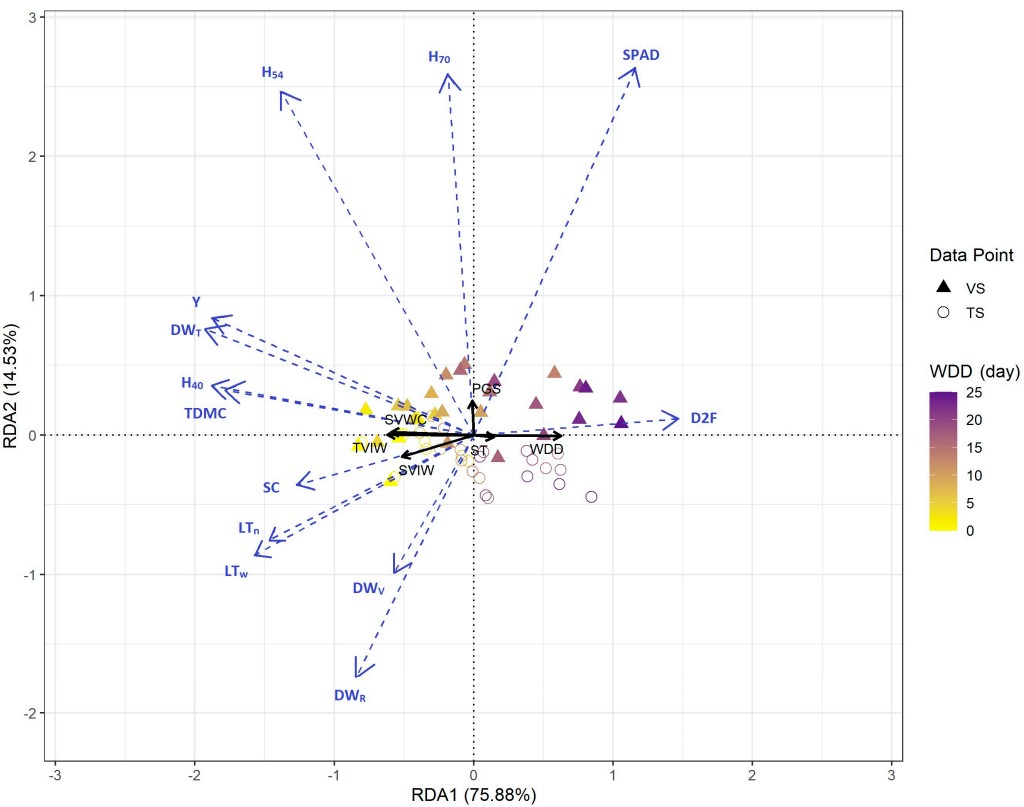

**Figure 4.** Redundancy analysis biplot. Water deficit treatment, soil condition and growth stage-related variables are used as independent variables (solid black vector lines), whereas plant growth, yield and tuber quality measures are used as dependent variables (dashed light blue vector lines). Data points are plotted in the same space (triangle symbol for vegetative growth stage and circle symbol for tuberization stage; colour reflects the water deficit duration). Eigenvalues are standardized to 1.000 and the cumulative percentage variance of each axis is shown in the following bracket. The first axis appears to represent the effects of water deficit, as the water deficit duration (WDD), total volume of water added (TVIW) and soil moisture (SVWC) all align well with this axis. The second axis appears to represent the effect of the growth stage, as the potato growth stage (PGS) variable aligns well with this axis and the data points for the two potato growth stages are mostly located on one side of the second axis (above for VS and below for TS).

Although the effects on vegetative growth, biomass production and tuber characteristics for the VS and TS treatments were mostly similar, there were some noticeable differences. First, plant height after the drying period recovered faster for VS-25 than TS-25 (Figure 2a) and the intercept values for the regressions between plant height and water

deficit duration were consistently greater for VS than TS (Table 1), both of which indicate that plant vegetative growth can recover better under VS than TS. However, better recovery of vegetative growth did not lead to better recovery of other plant growth and tuber characteristics measures. In fact, the slopes (absolute values) of the regression lines between water deficit duration and days to flowering and all biomass and tuber characteristics measures (except for tuber dry matter content) for VS were all slightly greater than those for TS, indicating that water deficit during VS had a slightly stronger impact on those parameters than water deficit during TS (Table 1). This emphasizes the long-lasting effects of water deficit in the earlier stages of plant development. One sign of such an effect is that for VS, the SPAD meter readings during the drying period were much greater than those for the control, whereas for TS, the differences were minimal (Figure 2b). It has been reported that the SPAD meter readings (greenness) increased when the potato plant was under water deficit, and such a greenness increase coincided with a leaf expansion cease [22]. In our study, although the SPAD meter reading recovered after the drying period ended under VS, it is possible that the leaf expansion did not recover well and with less leaf area, the primary production and biomass accumulation were both reduced. The difference between the two growth stages is also reflected in the RDA biplot in that: (1) $H_{54}$, $H_{70}$ and SPAD vectors are positive on axis 2, indicating that water deficit during VS had better recovery of plant growth and greener leaves; (2) $DW_V$ and $DW_R$ are negative on axis 2, indicating that water deficit during TS tended to accumulate more dry matter; and (3) other factors are not aligned well with axis 2, indicating that the effects of water deficit on them are similar for the two stages (Figure 4).

It should be noted that the differences between VS and TS described above with the water deficit duration did not seem to hold with the total volume of irrigation water (TVIW). The slopes (absolute values) of the regression lines for VS and TS between plant growth and tuber characteristics measures and TVIW did not have a consistent pattern (i.e., those for VS were not always greater than those for TS, Figure 3). This is probably due to the difference in the amount of irrigation water needed during the two stages. During the vegetative growth stage, the plant was not fully grown, so the evapotranspiration was low and thus the amount of irrigation water needed was low as well. This was clearly shown with the control pots, for which the TVIWs for the 25 days of water deficit periods for the VS treatment (8.9 L) were much lower than those for the TS treatment (14.2 L). With a lower need for water during the VS, the marginal benefits of adding water (i.e., the slopes of the regression lines) will be lower as well.

### 4.2. Implications for Supplement Irrigation Practices in Humid Climate Region

The results from this study indicate that for both the vegetative growth and tuberization stages, a periodic water deficit will negatively affect potato plant growth, yield and quality. The correlations between the water deficit duration and the plant growth and tuber characteristic measures were mostly linear and a switching point after a certain length of drying when the impact became significantly stronger or weaker was not found. Therefore, within each growth stage, the benefit of irrigation was the same for the whole period of the growth stage and there were no additional benefits for irrigation at a certain time. This implies that irrigation should be applied whenever there is a water deficit. Even with only a short water deficit duration (e.g., one day), irrigation will have some benefits and should not be skipped. On the other hand, the results imply that under water deficit conditions, any irrigation water added will help and since there is no difference in the timing of irrigation, irrigation should be applied whenever water is available.

Comparing the two growth stages examined, the effects of the length of the drying period were similar. Avoiding water deficit for the vegetative growth stage may be more important given the slightly stronger effects observed with the water deficit duration and the long-lasting effects of damages caused by water deficit in VS. This implies that irrigation should be applied as early as possible whenever a water deficit condition starts. This seems to contradict many previous studies on potato drought hardening or acclimation—a strategy

used to enhance potato's tolerance to water deficit in later stages by introducing the plant to mild water stress in an earlier stage [23–26]. The reason for this apparent contradiction lies in the fact that the drought hardening or acclimation strategy was developed for dry climate regions to reduce yield losses due to water deficit in the later growing stages. However, such benefits will not be realized when there is no water deficit in the later growing stages, as was the case in our experiment. Therefore, there is no actual contradiction between our study and those previous studies. In fact, EI-Abedin et al. [16] also found that deficit treatment and partial root-zone drying decreased potato fresh and dry weight compared to the full irrigation treatment, which is similar to our results.

Considering the climate conditions in Atlantic Canada and similar cold, humid climate regions, there is often excessive water in the spring when snow melts. Surface and subsurface drainage systems are used to drain the water out of the fields, usually directly into streams and rivers. The added water to the river system contributes to the heightened peak flow, which often leads to flooding in the early spring. If this water can be stored in the landscape (e.g., in ponds), it can be used later in the growing season for crop irrigation [27]. If the water is used on potatoes, based on a Canada-wide average farm potato price of CAD 302/Mg [28] and the results from this study, the economic return per cubic meter of water for potato yield increase alone can be calculated as CAD 4.9 and 5.8 for irrigation during the vegetative growth and tuberization stages, respectively. With the additional benefits of increasing tuber size, the economic return on irrigation can be greater. Moreover, when the excessive water in the spring is restored, there will be less water flowing into the river system, reducing the peak flow and thus the risk of flooding during that time. The economic return for peak flow reduction is difficult to assess but can be substantial in the future given that the frequency of flooding events is expected to rise due to climate change [4,29].

### *4.3. Future Studies*

As an extension of a previous pot study, this study was also based on a pot experiment with a single variety of Russet Burbank and, therefore, shares the same advantages and limitations as discussed in that paper [12]. A logical next step is to carry out some field studies to further examine how supplemental irrigation can be used to enhance potato yield and quality in a real-world setting. For example, potato varieties respond differently to water stress [30–32]. The cultivar tested in this study, Russet Burbank, has poor drought resistance. Therefore, similar experiments for other major varieties need to be carried out to quantify such differences. Also, this study used 70% of the available water content (AWC) and the permanent wilting point (PWP) as the controlled soil moisture levels to trigger irrigation for the no-water-stress and water-deficit conditions, respectively. However, the severity of the water deficit in the real world will vary. So, what if the soil moisture is at 50% of AWC or below PWP? How would different water deficit severity levels interact with the different durations and different potato growing stages? More studies are needed to answer these questions.

The climate conditions also need to be considered in the design of field studies. As mentioned above, in Atlantic Canada and similar cold, humid climate regions, there is often excess water in the spring. A water deficit is more likely to occur when crops start to grow and evapotranspiration increases [33]. So the overall strategy for water management should be saving water in the spring and using it in the summer. However, with climate change, extreme weather conditions become more frequent and we may see a water deficit occurring in the early crop growing stages, such as the sprouting and vegetative growth stages for potatoes. Given the long-lasting effects of water deficit in the early growing stages, the linear relationship between yield and tuber quality parameters and the water deficit duration observed in this study, when water deficit does occur during these growing stages, it would be beneficial to use the water right away rather than saving it for later. However, more studies are needed to determine when to trigger irrigation.

To better manage water use in agricultural landscapes, Li [27] proposed a Landscape Integrated Soil and Water Conservation (LISWC) system to utilize the hydrological difference in different slope positions in the landscape to drain the excessive water in the spring via surface and subsurface drainage systems from the upslope areas and store the water in a retention structure in the depression or bottom of slope areas, using the water for supplemental irrigation later in the growing season. In such an LISWC system, the temporal difference in water supply (water excess in the spring and water deficit in the summer) is balanced by the spatial difference in hydrology (water draining from the upslope area and storage in the depression or bottom slope areas). The LISWC system also has the potential to reduce surface runoff, water erosion and nutrient leaching and, therefore, reduce the environmental impact of agriculture. A field study is needed to quantify how an LISWC system can be used most efficiently to achieve both crop yield and environmental protection goals.

## 5. Conclusions

The pot experiment conducted in this study confirms that a water deficit negatively affects potato plant growth, yield and tuber quality. The above-ground vegetative growth will mostly recover after the water-deficit period ends, especially when the water-deficit period is early in the growing season. However, the impacts of water deficit on primary production and biomass accumulation are long lasting, especially with early-season water deficits. In the end, the impacts of the water deficit duration for the vegetative growth stage are very close or even slightly stronger than those for the tuberization stage.

Overall, for both the vegetative growth and tuberization stages, a longer water deficit duration may not significantly change the final plant height but will significantly delay flowering and reduce tuber dry weight, total biomass, yield, tuber dry matter content and the share of large tubers. Moreover, the impacts of the water deficit duration appear to be linear, indicating that the effect of irrigation will be similar regardless of the timing. The regression analyses suggest that with a prolonged water deficit per day, there will be a yield loss of 3.1% and 3.4% for vegetative growth and tuberization, respectively. Similarly, per liter of irrigation water, there will be an increase in yield of 16.3 g and 19.1 g for vegetative growth and tuberization, respectively.

**Supplementary Materials:** The following supporting information can be downloaded at: https://www.mdpi.com/article/10.3390/agriculture13102007/s1, Table S1: Correlation coefficient between measured parameters for the vegetative growth and tuberization stage (* $p \leq 0.05$; ** $p \leq 0.01$; *** $p \leq 0.001$).

**Author Contributions:** Conceptualization, S.L., Y.K., C.W. and S.H.; methodology, S.L., Y.K., S.H.; software, C.W. and F.Z.; validation, S.L.; formal analysis, S.L., F.Z. and C.W.; investigation, Y.K. and S.L.; resources, S.L.; data curation, Y.K.; writing—original draft preparation, S.L.; writing—review and editing, S.L., Y.K. and C.W.; visualization, S.L. and F.Z.; supervision, S.L.; project administration, S.L.; funding acquisition, S.L. All authors have read and agreed to the published version of the manuscript.

**Funding:** This research was funded by Agriculture and Agri-Food Canada (AAFC) through an A-base project "Landscape Integrated Soil and Water Conservation (LISWC) on sloping fields under potato production in Atlantic Canada" (J-001754, PI: Li) and an Agri-Science project "Regenerative agriculture mitigating soil degradation and climate change challenges" (J-002703, PI: Goyer) and through an Enabling Agricultural Research and Innovation (EARI) project "Modeling the effects of water stress and supplemental irrigation on potato production in NB under climate change" (J-002472, PI: Li) managed by the province of New Brunswick via the Canadian Agricultural Partnership.

**Institutional Review Board Statement:** Not applicable.

**Data Availability Statement:** Data archive are not available.

**Acknowledgments:** The authors would like to thank Angela McMillan and Zane Jeffries for technical assistance, John Gillan and the greenhouse crew for managing the greenhouse for this experiment.

**Conflicts of Interest:** The authors declare no conflict of interest.

## Nomenclature

| | |
|---|---|
| AWC | Plant available water capacity (%) |
| D2F | Days to flowering (day) |
| DAP | Days after planting (day) |
| CIWV | Cumulative Irrigation Water Volume (L) |
| DIWV | Daily Irrigation Water Volume (mL) |
| $DW_R$ | Roots dry weight (g) |
| $DW_T$ | Tuber dry weight (g) |
| $DW_V$ | Vegetation dry weight (g) |
| FC | Field capacity (%) |
| $H_{40}$ | 40-day plant height (cm) |
| $H_{54}$ | 54-day plant height (cm) |
| $H_{70}$ | 70-day plant height (cm) |
| IWUE | Irrigation water use efficiency (g $L^{-1}$) |
| $LT_n$ | Share of large tubers (>35 mm) by fresh weight (%) |
| $LT_w$ | Share of large tubers (>35 mm) by tuber number (%) |
| PGS | Potato growth stage |
| PWP | Permanent wilting point (%) |
| SC | Stomatal conductance (mmol $m^{-2}$ $s^{-1}$) |
| SPAD | SPAD meter reading |
| ST | Soil Temperature (°C) |
| SVWC | Soil volumetric water content (%) |
| TBM | Total biomass (g) |
| TDMC | Tuber dry matter content (%) |
| TVIW | Total volume of irrigation water (L) |
| SVIW | Volume of irrigation water during the growth stage of interest (L) |
| TS | Tuberization stage |
| VS | Vegetative growth stage |
| WDD | Water deficit duration (day) |
| Y | Potato yield (tuber fresh weight) (g) |

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
