# Peer review of "Water Deficit Duration Affects Potato Plant Growth, Yield and Tuber Quality"

_agriculture, doi:10.3390/agriculture13102007_

Round 1

Reviewer 1 Report

 The authors did a lot of works on the effects of water deficit duration on potato plant growth, yield and tuber quality under potted control conditions. The authors selected two water sensitive periods of potatoes for quantitative research, and the research results have good innovation and inspiration. It has certain guiding significance for exploring supplemental irrigation to mitigate the impact of water deficit on potato production in future production at humid climate regions.

  Generally, the manuscript is quite interesting and well organized. I do believe the topic is relevant in view of the economical and nutritional importance of tuber crops (including sweet potatoes and yams) worldwide. I think this manuscript is suitable for publication.

The main concern that I have is that the reported work was done in pot experiment with one variety. The potato variety chosen is drought tolerant or water-insensitive variety? The conclusions of the experiment will vary greatly.

I enjoyed reading the manuscript and it is well referenced. It is generally well written though I feel there is a need to increase clarity of the methods. line 111 “The soil was sterilized and sieved through a 5 mm sieve.” What is the purpose of the author's sterilization, as the experimental treatment does not involve changes in microbial flora? Usually, soil can be sieved through a 2mm sieve, why use a 5mm sieve?

Suggest replacing rain-fed agriculture with potato in the key words section.

Best regards!

Author Response

Agriculture-2646259-R1 Reviewer #1

Comments and Suggestions for Authors

The authors did a lot of works on the effects of water deficit duration on potato plant growth, yield and tuber quality under potted control conditions. The authors selected two water sensitive periods of potatoes for quantitative research, and the research results have good innovation and inspiration. It has certain guiding significance for exploring supplemental irrigation to mitigate the impact of water deficit on potato production in future production at humid climate regions.

Generally, the manuscript is quite interesting and well organized. I do believe the topic is relevant in view of the economical and nutritional importance of tuber crops (including sweet potatoes and yams) worldwide. I think this manuscript is suitable for publication.

A: Thanks for the positive feedback!

The main concern that I have is that the reported work was done in pot experiment with one variety. The potato variety chosen is drought tolerant or water-insensitive variety? The conclusions of the experiment will vary greatly.

A: This manuscript reports a follow-up study to a previous one, which has been published (12. Wagg et al., 2021). The advantages and disadvantages of pot experiment and the limitations of only one variety being tested have been discussed in much detail in that paper. In the future study section, we cited the previous study and discussed what questions can be answered with a field study. We revised this section and added some discussion for the impact of variety.

I enjoyed reading the manuscript and it is well referenced. It is generally well written though I feel there is a need to increase clarity of the methods. line 111 “The soil was sterilized and sieved through a 5 mm sieve.” What is the purpose of the author's sterilization, as the experimental treatment does not involve changes in microbial flora? Usually, soil can be sieved through a 2mm sieve, why use a 5mm sieve?

 A: Our centre has a focus on potato research and have researchers working on different aspects of potato such as potato variety tests, disease and nutrient related experiments. All soil must be sterilized before it can be used in our greenhouse to prevent soil borne pests and diseases. The 5mm sieve is used rather than 2mm sieve because: 1) the efficiency for soil processing is much higher with 5mm sieves; and 2) the 5mm sieve keeps the larger soil aggregates so that the soil is more similar to field soil.

Suggest replacing rain-fed agriculture with potato in the key words section.

A: We added the variety “Russet Burbank” onto the list of key words. We kept the rain-fed agriculture because it sets the context of water stress, which is quite different from the arid and semi-arid areas where most potato water stress studies have been conducted.

Reviewer 2 Report

Comments in file below

Author Response

Agriculture-2646259-R1 Reviewer #2

Comments and Suggestions for Authors

The manuscript submitted to me for review concerns the impact of drought stress in two developmental stages of potato on plant development, tuber yield and its quality. The work is interesting and expands existing knowledge on this topic. So far, most of the work on this issue concerned the impact of drought during tuberization, which is the most sensitive time of the plant to water shortage. The presented work shows that the vegetative growth period is an equally or perhaps even more sensitive phase. The presented relationships may have great application in agricultural practice.

A: Thanks for the positive feedback!

My comments concern the following issues:

  1. In the Introduction, the authors write that there is only one work on the impact of drought at various stages of the potato plant. To my knowledge there are more.

A: Thanks for pointing this out! We revised the manuscript accordingly and added some references to this.

  1. Methodology - I believe that too few plants were tested. After the plant in combination with the applied drought is not enough. If 1 plant dies for various reasons and data is missing. There should be at least 2 plants each. I realize that it would greatly expand the experiment, but the results would be more reliable.

A: We would love to have more pots but as the reviewer noted, it doubles the work load, which is beyond what we can afford. An important aspect for this study is that the stats is mostly based on regression analysis, for which repeats are not needed. There are a total of 26 data points in the analysis. Missing a few data points should not affect the overall results much. 

  1. Research results. Most of the results are consistent with those previously presented in the literature. The only doubt is that the drought applied during the tuberization period did not significantly affect the above-ground plant mass and root mass. Previous research shows such relationships.

A: In fact, our previous study (12. Wagg et al., 2021) did find that drought during tuberization period will significantly affect both the above-ground and root biomass. However, this study is about the drought duration, not whether or not there is a drought. For both the vegetative growth and tuberization stages, the above-ground and root biomass data were highly variable. This high variability was related to the difficulties in sample handling and processing. The insignificance tests were likely a result of the high data variability (R2 values for the vegetative growth stage were low as well).

  1. The discussion is carried out correctly in relation to the existing literature in this field.

A: Thanks for the positive feedback!

  1. Substantive conclusions, properly formulated

A: Thanks for the positive feedback!

  1. Literature - too few items, lack of the latest works on the problems discussed.

A: We added more references as suggested.